# New Facets of DNA Double Strand Break Repair: Radiation Dose as Key Determinant of HR versus c-NHEJ Engagement

**DOI:** 10.3390/ijms241914956

**Published:** 2023-10-06

**Authors:** Emil Mladenov, Veronika Mladenova, Martin Stuschke, George Iliakis

**Affiliations:** 1Division of Experimental Radiation Biology, Department of Radiation Therapy, University Hospital Essen, University of Duisburg-Essen, 45122 Essen, Germany; veronika.mladenova@uk-essen.de (V.M.); martin.stuschke@uk-essen.de (M.S.); 2Institute of Medical Radiation Biology, University Hospital Essen, University of Duisburg-Essen, 45122 Essen, Germany; 3German Cancer Consortium (DKTK), Partner Site University Hospital Essen, 45147 Essen, Germany; 4German Cancer Research Center (DKFZ), 69120 Heidelberg, Germany

**Keywords:** DNA double strand breaks (DSBs), ionizing radiation (IR), homologous recombination (HR), RAD51, cancer therapy, radiation therapy

## Abstract

Radiation therapy is an essential component of present-day cancer management, utilizing ionizing radiation (IR) of different modalities to mitigate cancer progression. IR functions by generating ionizations in cells that induce a plethora of DNA lesions. The most detrimental among them are the DNA double strand breaks (DSBs). In the course of evolution, cells of higher eukaryotes have evolved four major DSB repair pathways: classical non-homologous end joining (c-NHEJ), homologous recombination (HR), alternative end-joining (alt-EJ), and single strand annealing (SSA). These mechanistically distinct repair pathways have different cell cycle- and homology-dependencies but, surprisingly, they operate with widely different fidelity and kinetics and therefore contribute unequally to cell survival and genome maintenance. It is therefore reasonable to anticipate tight regulation and coordination in the engagement of these DSB repair pathway to achieve the maximum possible genomic stability. Here, we provide a state-of-the-art review of the accumulated knowledge on the molecular mechanisms underpinning these repair pathways, with emphasis on c-NHEJ and HR. We discuss factors and processes that have recently come to the fore. We outline mechanisms steering DSB repair pathway choice throughout the cell cycle, and highlight the critical role of DNA end resection in this process. Most importantly, however, we point out the strong preference for HR at low DSB loads, and thus low IR doses, for cells irradiated in the G_2_-phase of the cell cycle. We further explore the molecular underpinnings of transitions from high fidelity to low fidelity error-prone repair pathways and analyze the coordination and consequences of this transition on cell viability and genomic stability. Finally, we elaborate on how these advances may help in the development of improved cancer treatment protocols in radiation therapy.

## 1. Introduction

In addition to surgery and chemotherapy, radiation therapy is a key regimen in modern cancer treatment that utilizes various ionizing radiation (IR) modalities to mitigate cancer progression. IR generates ionizations and ionization clusters that hold enough energy to evict electrons from water or DNA, thus generating a plethora of DNA lesions with a wide spectrum of complexity [1]. The most detrimental DNA lesion that is induced by IR is the double strand break (DSB). IR-induced DSBs contribute considerably to the efficacy of radiotherapy applications [2,3,4]. If left unrepaired or incorrectly repaired, DSBs result in enhanced levels of chromosomal abnormalities that are associated with increased cell lethality, and are also considered a hallmark of cancer development [5,6,7,8].

In cells of higher eukaryotes, four DSB repair pathways operate in concert to counteract the adverse effects of IR. Indeed, classical non-homologous end joining (c-NHEJ), homologous recombination (HR), alternative end joining (alt-EJ), and single strand annealing (SSA) function to restore genome integrity and DNA sequence around the break [3,9,10]. Notably, DSBs are frequently repaired inaccurately by these repair pathways, leading to cell death or mutagenesis. Recent studies suggest that “processing accidents” directly caused by the engagement of a specific DSB repair pathway underpin many of the adverse effects of DSBs on cell survival or the induction of genomic alterations [3,4]. This perspective challenges the notion that the four DSB repair pathways represent equivalent options for repair and highlights the importance of better understanding the evolutionary significance of their mechanistic diversity and their considerable differences in accuracy, efficiency, and speed.

There is evidence that the upregulation of certain DSB repair pathways in the genomically altered background of cancer cells can increase their radioresistance [11,12,13,14]. Consequently, several studies have explored means to improve radiotherapy by antagonizing key DSB processing factors, thereby suppressing DSB repair or shifting the balance between DSB repair pathways [14,15,16,17]. A widely accepted premise in the field is that a mechanistic understanding of the molecular principles driving the hierarchical coordination in the engagement of DSB repair pathways will highlight powerful means to improve radiotherapy.

## 2. Molecular Mechanisms of DSB Repair in Higher Eukaryotes

DNA DSBs are endogenously induced randomly in the genomes of higher eukaryotes as a result of increased oxidative stress or accidents during DNA replication [18,19]. Despite the potentially detrimental nature of DSBs, their site-specific, enzymatic induction is programmed in defined cell types of the immune system and serves essential functions, e.g., in V(D)J and class switch recombination (CSR) [20,21,22]. Moreover, Spo11-mediated DSBs are introduced at specific locations in germ cells to induce meiotic recombination between two homologous chromosomes [23]. In addition, recent studies suggest that DSBs occur as intermediates during neurobiological processes related to memory development [24,25,26]. From an evolutionary perspective, DSBs are also thought to provide opportunities for genetic rearrangements and the integration of DNA fragments into the genome [27]. Therefore, it is not surprising that vertebrate cells are equipped with a sophisticated DSB repair network that forms an integral part of the general DNA metabolism; it operates as a competent guardian of the genome against endogenous and exogenous insults but also facilitates the above mentioned essential biological processes.

Multiple chemical and physical agents interact with DNA and generate DSBs, often almost randomly. From the plethora of such agents, IR takes a special place. It has been documented that even a single unrepaired DSB can trigger cell death. Therefore, the potential of DSBs to induce cell death is exploited in radiation therapy that utilizes IR to target the viability of cancer cells [12,28].

Innately, cancer cells encounter a higher endogenous DSB burden in comparison with normal cells, owing to oncogene-induced replication stress, increased metabolic activity, and acquired defects in major DSB repair, as well asDNA damage response (DDR) mechanisms [29,30]—all components of the carcinogenesis. As a consequence of this, hyper-proliferating cancer cells are forced to rely on altered pathway balance in their DSB repair pathway network that may include the suppression of some repair pathways and the promotion of the others. Such shifts frequently alter the cancer cell response to radiation, which may facilitate or hamper eradication during radiation therapy [31].

Inherited DNA repair defects can predispose an individual to develop certain types of cancer and the very same susceptibility may be therapeutically exploited to preferentially eradicate cancer cells in tumors [32,33]. Therefore, a molecular understanding of DSB repair mechanisms and their control are important prerequisites for the development of improved radiotherapy treatment protocols.

The four DSB repair pathways mentioned above have distinct cell cycle and DNA sequence homology dependencies and operate with different fidelity and kinetics throughout the cell cycle [4,19,34,35]. The engagement of a particular DSB repair pathway is predicated, to a large extent, on DNA end resection, a process of 5′- to 3′- degradation of one DNA strand to generate a single strand DNA-overhang with a free 3′-OH terminus. These single strand DNA intermediates are essential entities that trigger homology-directed repair mechanisms.

As a consequence, DSB repair pathways are frequently classified as resection dependent (HR, alt-EJ, and SSA) and resection independent (c-NHEJ) (Figure 1). DNA end resection-dependent DSB repair pathways are characterized by a pronounced cell cycle dependence, which may be harnessed to increase the efficacy of radiation therapy (Figure 1). In contrast, c-NHEJ is renowned for being active throughout the entire cell cycle and there are no major documented differences in the availability, accessibility, and efficiency of its integral components at different cell cycle stages (Figure 1) [9,36]. Notably, DNA end resection is reduced in the G_1_- and especially in the G_0_-phase of the cell cycle, which renders c-NHEJ essential in quiescent and terminally differentiated cells. Although, there is evidence that limited DNA end resection occurs in G_0_- and G_1_-cells, and that this could promote a form of c-NHEJ that relies on resection to operate [37,38,39], it is clear that resection is actively suppressed in pre-replicative cell cycle phases by a complex regulatory network acting on DNA end resection activities, and regulating their expression and ubiquitination-mediated degradation [40,41,42,43]. Therefore, DNA end resection-dependent DSB repair pathways generally gain activity during S- and G_2_-phase of the cell cycle (Figure 1).

Both initial and long-range DNA end resection are extremely important for the initiation of HR and SSA, as the formation of sufficiently long 3′-single strand DNA overhang intermediates is essential for the initiation of homology-directed DSB repair [44,45]. However, it remains open whether alt-EJ requires DNA end resection, and whether certain alt-EJ sub-pathways are resection independent. Recent evidence indicates that long-range resection suppresses alt-EJ repair, while short-range resection promotes its activity [46,47]. Moreover, pioneer studies report that, under certain conditions, alt-EJ could engage in the repair of non-resected DSBs or DSBs that undergo limited DNA end processing [48,49,50,51]. These observations emphasize the plasticity of this repair pathway, a property that endows it with interesting “back-up” functions, whenever other repair pathways engage first at a DSB but somehow fail to completely restore DNA integrity. Consistently, it was shown that alt-EJ activity is detectable during the G_1_-phase of the cell cycle and that it becomes enhanced in the G_2_-phase, where the abundance of end resection factors is at a maximum [52,53]. In line with these results, our studies also indicate that alt-EJ is strongly suppressed in the stationary phase of growth where cells are predominantly in the G_0_-phase [54,55].

The accumulated data on DSB repair pathway activity in mitosis remain controversial (Figure 1). There are reports indicating no active DSB repair in mitosis [56,57], while other reports highlight the role for HR, SSA, and c-NHEJ [58,59]. Recent results have identified the 9-1-1 complex (RAD9A-HUS1-RAD1) and its interacting partner, RHINO, as critical factors of microhomology mediated end joining (MMEJ) during mitosis [60]. RHINO has been shown to play an essential role during the mutagenic alt-EJ repair in mitosis by interacting with polymerase theta (POLθ) and facilitating its recruitment to DSBs [60]. Our current work also shows that POLθ contributes to mitotic DSB repair and rejoins a fraction of DSBs in the absence of DNA-dependent protein kinase catalytic subunit (DNA-PKcs), an essential factor of c-NHEJ (unpublished results).

Importantly, the functional interplay between major DSB repair pathways, in combination with their cell cycle requirements, offers therapeutic opportunities to selectively target repair-compromised tumors, based on the concept of synthetic lethality [28,61,62]. In the following section, we will outline the basic principles of DSB repair processes identified in higher eukaryotes, with emphasis on newly characterized factors involved in c-NHEJ and HR. We will also outline the main control mechanisms utilized by cells to modulate the activity of certain repair mechanisms and therefore increase their survival.

### 2.1. Classical Non-Homologous end Joining (c-NHEJ)

The major pathway implicated in DSB repair in both actively proliferating and quiescent cells is c-NHEJ (Figure 1). The efficiency of c-NHEJ remains relatively constant throughout the cell cycle and is essential for the elimination of DSBs in the majority of cells, particularly when they enter quiescence or become terminally differentiated [63,64]. C-NHEJ is initiated by the binding of KU70/80 heterodimer to DSB ends, followed by the recruitment of DNA-PKcs, to form an active DNA-PK holoenzyme [65,66,67,68].

DNA-PKcs, a member of the PIKK family of protein kinases, has been originally discovered in human cells as a protein kinase that is stimulated after interaction with double-stranded DNA. DNA-PKcs is absent in bacteria and many lower eukaryotes, and the prevailing assumption was that it is a vertebrate-specific PIKK [69]. However, a recent study reports that protein sequences and structural motifs, similar to those of DNA-PKcs, have been detected in invertebrates, fungi, plants, and protists [69].

The DNA-PK holoenzyme is reported to serve as a landing pad for the recruitment of other c-NHEJ factors. A commonly accepted model is that, prior to DNA ligation, DNA-PKcs must be displaced from DNA ends by autophosphorylation to free them up for ligation [70]. However, new data suggest that DNA-PKcs could remain bound to DNA ends and help coordinate the overall repair reaction including the transfer of DNA ends for processing by HR [71].

The generation of ligation-compatible DNA ends requires DNA end processing, which can include enzymatic modification, excision, or addition of nucleotides. This is particularly important for IR-induced DNA ends that contain damaged nucleotides and are typically incompatible for direct ligation [72,73]. Activities contributing to DNA end processing include the Artemis nuclease, the polymerases of the POL X family, POL μ and POL λ, the tyrosyl-DNA phosphodiesterase 1 (TDP1), and the polynucleotide kinase 3′-phosphatase (PNKP) [74]. The chemistry of the DNA end determines which of these factors are required for the preparation of ligatable DNA ends.

Ligation of the broken DNA termini is catalyzed by the DNA Ligase 4, X-ray cross complementing protein 4 (LIG4/XRCC4) complex, whose efficiency is facilitated by the XRCC4-like factor (XLF), and/or by the paralogue of XRCC4 and XLF (PAXX) [63]. It has been shown that PAXX acts as a stabilizing molecule for the main NHEJ components but could also redirect DNA end processive enzymes to the DSB [75].

Recent studies identified new proteins that are important for the c-NHEJ function, such as the modulator of retroviral infection (MRI)/cell cycle regulator of NHEJ (CYREN) [76,77], the transactivation response DNA binding protein (TARDBP) of 43 kDa (TDP-43), RNase H2, and the intermediate filament family orphan (IFFO1) protein [66]. There is evidence that MRI/CYREN plays a dual role in DSB repair by stimulating c-NHEJ in the G_1_-phase, while suppressing c-NHEJ efficiency in the S- and G_2_-phases [66,78]. TDP-43 has been shown to recruit LIG4/XRCC4 complex to DSB sites and to stimulate ligation in neuronal cells, while RNase H2 has been revealed as the ribonuclease that excises ribonucleotides inserted during DNA synthesis by DNA POLμ/TdT [66]. IFFO1 is a nuclear matrix-associated protein, involved in the immobilization of broken DNA ends and the suppression of structural chromosomal abnormalities (SCAs) formation during DSB repair by c-NHEJ [79]. This protein also acts as a scaffold mediating the XRCC4/LMNA complex formation involved in the immobilization of the broken DNA ends to the nuclear lamina [79].

In addition to the above factors, the C2H2-type ZnF protein, ZNF384, has been described as an important DSB repair component essential for the repair of DSBs by c-NHEJ [80]. During the initial steps of c-NHEJ repair, PARP-1/PAR-mediated chromatin expansion initiates the recruitment of ZNF384 to the sites of DNA damage, which catalyzes the physical interactions between the N-terminus of ZNF384 and KU heterodimer. The interaction of ZNF384 with both DNA and KU70/KU80 is critical for the processive loading of KU and the subsequent recruitment of DNA-PKcs [80].

Recently, a new factor stimulating the recovery from an IR-induced G_2_-checkpoint was found to promote DSB repair by c-NHEJ [81]. PHD finger protein 6 (PHF6) is encoded by a gene mutated in Börjeson–Forssman–Lehmann syndrome and many leukemic cancers. It is rapidly recruited to the sites of DSBs in a PARP-dependent manner and its recruitment is essential for the efficient function of c-NHEJ [81]. The number of essential factors involved in DSB repair by c-NHEJ continuously increase, suggesting that c-NHEJ is endowed with an elaborate, highly coordinated protein apparatus that is well-integrated in the overall machinery regulating the overall DSB repair program and pathway choice.

### 2.2. Homologous Recombination (HR)

HR is the only DSB repair pathway that results, owing to its templated function (sister chromatid homology), in the faithful restoration of both DNA integrity and the original DNA sequence at the DSB [82,83]. HR can accommodate a wide spectrum of structural DNA end configurations, such as variations in the 3’-overhang length, DNA end sequence permutations, and DNA end chemistry. HR is an elaborate process, which strongly depends on homology. Therefore, its activity is restricted to the late S- and G_2_-phase of the cell cycle, where the homologous sister chromatid is present [82,83] (Figure 1). HR is suppressed in the G_1_-phase, where only homologous chromosomes are available. It is speculated that the HR preference for the sister chromatid as substrate is founded on its proximity, which is ensured by proteins such as condensins, cohesins, and to some extent, CTCF [84,85,86]. There is also evidence that unscheduled HR may be initiated in the G_1_-phase, which in many situations results in the formation of abortive recombination intermediates and the accumulation of genomic alterations [87].

HR is divided into three major stages: pre-synaptic, synaptic, and post-synaptic stage. The initial step of HR is characterized by the MRN (MRE11-RAD50-NBS1)-mediated detection of the DSB, followed by the recruitment of the CtBP-interacting protein (CtIP) (Figure 2) [88,89]. Initially, CtIP generates a nick in one of the DNA strands near the 5′- end and initiates MRE11 dependent 3′- to 5′-nucleolytic degradation to create short single strand 3′-overhangs [85,86]. This short-range resection is followed by long-range resection mediated by the recruitment of multiple nucleases like Bloom syndrome helicase/DNA replication ATP-dependent helicase/nuclease 2 (BLM/DNA2) complex, and exonuclease 1 (EXO1) (Figure 2). In addition to the BLM/DNA2 complex, recent studies implicate the activities of Werner syndrome helicase (WRN) and RECQL helicase in the resection of degraded nascent DNA strands during the restart of blocked or collapsed replication forks [90,91]. These findings indicate that the above helicases may play a role in radiation-induced long-range resection, as well.

Long-range resection forms an extended tract of single strand DNA with a regular length of hundreds to couple of thousands nucleotides [92]. DNA end resection is a decisive point in the regulation of DSB repair pathway choice [92]; therefore, its regulation will be discussed in detail in the next sections.

The single strand 3′-tails are rapidly covered by the replication protein A (RPA) heterotrimeric complex, which protects the single strand DNA from nucleolytic degradation and mitigates the formation of secondary structures [93,94,95]. In the following stages, the BRCA2 mediator protein, in coordination with PALB2 and the BRCA1/BARD1 complex, displaces RPA from ssDNA and facilitates the loading of the RAD51 recombinase, a central and essential activity of HR. The formation of RAD51 nucleoprotein filament marks the pre-synaptic stage of HR. This process is additionally facilitated by the group of proteins indicated as RAD51 paralogs (RAD51B, RAD51C, RAD51D, XRCC2, and XRCC3) [96,97,98] and is also promoted by the RAD54 and RAD54B proteins [99,100,101].

During the synaptic stage of HR, the RAD51 nucleoprotein filament initiates a homology search in the sister chromatid. When it is found, it invades the double strand DNA to form a Holliday junction—a key intermediate in HR [102]. Subsequently, RAD54 functions to promote DNA synthesis and branch migration by dislocating RAD51 from heteroduplex DNA. In the post-synaptic steps, the extended Holliday junction is resolved in a variety of ways that define specific HR sub-pathways [103]. It is believed that the synthesis-dependent strand annealing (SDSA) is the most relevant sub-pathway in the repair of IR-induced DSBs by HR. In this final step, the newly synthesized DNA strand anneals with the similarly processed second DNA strand to restore integrity in the molecule, and the process is completed by DNA synthesis and ligation [104].

Multiple novel HR-related factors have been identified. This new group of proteins expands the mechanistic understanding of DSB repair by HR and provides a new layer of complexity that is thought to fine-tune HR. Thus, topoisomerase 1-binding arginine/serine-rich protein (TOPORS) [105], nuclear ubiquitous casein kinase and cyclin-dependent kinase substrate 1 (NUCKS1) [106], RAD51-associated protein 1 (RAD51AP1) [107], HOP2-MND1 heterodimer [108], the SWI5-MEI5 (C9orf119-C10orf78) complex [109], RAD51-binding protein fidgetin-like 1 (FIGNL1) [110,111], the SWS1–SWSAP1-SPIDR multimer complex [112], and the product of the DEK proto-oncogene [113,114] have been reported to directly modulate and positively regulate HR in eukaryotic cells.

In the same context, TOPORS has been found to mediate the SUMOylation of RAD51 at L57 and L70, which is critical for the association of RAD51 to BRCA2 and the formation of RAD51 nucleoprotein filament [105]. On the other hand, NUCKS1 is a novel RAD51AP1 paralog, which stimulates the ATPase activity of RAD54 and the RAD51–RAD54-mediated strand invasion step during HR [106]. The HOP2-MND1 heterodimer promotes the DNA strand exchange activities of RAD51 and helps to load RAD51 on single strand DNA by restraining its double strand DNA-binding activity. However, during the homology search, HOP2-MND1 switches gears and facilitates the binding of RAD51 to double strand DNA [108]. SWI5-MEI5, FIGNL1, and DEK have been found to directly interact with RAD51 and possibly to mediate and stabilize the formation of the RAD51 nucleoprotein filament.

Recently, PTEN has also surfaced as a factor whose deficiency severely impairs RAD51 foci formation, as well as the efficiency of HR [115,116]. Moreover, along with the above presented studies, there is a report identifying HR-related factors by a high-throughput assay [117]. However, for the majority of the above HR proteins, the mode of action is elusive, as there are no clearly defined molecular mechanisms for their functions in HR. Nevertheless, the identification of such proteins may turn out to be important for understanding the overall control of the DSB repair pathway choice.

### 2.3. Alternative End-Joining (Alt-EJ)

When c-NHEJ or/and HR are compromised, or when accidents in DSB processing by c-NHEJ or HR occur that compromise ultimate repair, an alternative form of DSB repair comes to the fore, alt-EJ. Alt-EJ, also known as backup non-homologous end joining (B-NHEJ) [118,119], microhomology-mediated end joining (MMEJ) [120], or theta-mediated end joining (TMEJ) [46,121], often benefits from short (2 to 20 nt) stretches of microhomology that are exposed following limited processing of DSB ends. The accumulated evidence on the topic suggests that the term alt-EJ refers to different subpathways with central proteins and engagement characteristics that remain to be fully elucidated. In the following paragraphs, the roles of proteins implicated in alt-EJ are discussed without an attempt to systematically organize them in sub-pathways. However, there are excellent reviews on the topic [122,123,124,125,126].

It is thought that an initial step in alt-EJ is DSB recognition by PARP-1 [127,128] that may promote short-range DNA end resection by CtIP and the MRN complex. Repair may continue with the annealing of 2–20-nt (most often 3–8-nt) microhomologies in 3′-tails, and may be facilitated by DNA polymerase θ (POLθ); unpaired non-homologous 3′-tails are digested by the ERCC1/XPF nuclease (Figure 1). Gaps within the DNA strands may be filled-in by POLθ-mediated DNA synthesis [129], and DSB ends are rejoined by the DNA Ligase 3 (LIG3)/XRCC1 complex [130,131]. In the absence of the more efficient LIG3, DNA Ligase 1 (LIG1) can also take over to catalyze the final step of DNA ligation [132].

Alt-EJ operates with slower kinetics in all phases of the cell cycle, but its activity peaks in the G_2_-phase and is dramatically reduced in the stationary phase of growth when cells enter the G_0_-phase [54] (Figure 1). Alt-EJ operates with lower efficiency than c-NHEJ and is considered markedly error-prone. Thus, deletions and other modifications at the junction occur more frequently when compared with c-NHEJ. It is also relevant that during the processing of DSBs by alt-EJ, the joining probability of unrelated DNA ends is markedly increased. Consistently, alt-EJ is a dominant source of SCAs formation [133].

### 2.4. Single Strand Annealing (SSA)

Single strand annealing is a RAD51-independent homology-dependent mechanism that bridges two homologous 3′-single-stranded DNA ends at tandem repeated regions, resulting in the obligate deletion of the fragment between the repeats [134,135,136]. SSA requires extensive DNA end resection and RPA displacement to reveal complementary homologous sequences. SSA relies on RAD52 activity for the annealing step, the structure-specific endonuclease XPF–ERCC1 for the removal of unpaired non-complementary tails, and LIG1 for ligation of the remaining nick (Figure 1) [135]. SSA is inherently mutagenic and results in the loss of multiple kilobases of genetic information [136]. Therefore, it can be detrimental for the genome when it takes place in gene-rich regions but may be tolerated, and indeed enhanced, in the repetitive regions of the genome. Owing to the strong requirement of DNA end resection activities, SSA is probably active exclusively in the S- and G_2_-phase of the cell cycle.

## 3. Determinants of DSB Repair Pathway Choice: Emphasis on Resection

The evolution of the above described mechanistically distinct DSB repair pathways of varying fidelity suggests that, for the removal of all types of DSBs induced by IR at random genome locations, throughout the cell cycle, an array of multiple repair pathways is required. We postulate that optimal genome maintenance is best served when the repair pathway choice is programmed for priority engagement of the highest fidelity repair pathway available. Only when this engagement fails, or turns unproductive, lower fidelity repair pathways engage to still ensure the removal of the DSB. In this case, the associated infrequent errors are accepted as a compromise to the alternative of leaving the DSB unrepaired. In the application of this logic, it is always necessary to consider that the availability and relative activity of the different repair pathway changes throughout the cell cycle (see above). Indeed, all resection-dependent repair pathways become more active as the cell moves from the G_1_- to G_2_-phase, in contrast to c-NHEJ, which retains similar activity throughout the cell cycle [9,36,54,55].

In this framework of repair pathway choice, it is not necessary to postulate the induction of irreparable DSBs, as has frequently been done in the past in radiation biology to explain cell death and carcinogenesis. Instead, it can be postulated that DSBs (actually the vast majority of them) will be handled according to the above outlined logic, in a cell autonomous way, with the chemical and biological characteristics of the DSB and the available DSB repair pathways, deciding the ultimate repair pathway that will remove this particular DSB from the genome. Lethal or carcinogenic effects are, in this scenario, the rare byproducts of the inherent deficiencies of the repair pathway ultimately utilized. Indeed, SCAs, which are highly consequential at all endpoints, derive mainly from the engagement of c-NHEJ and alt-EJ at particular DNA lesions, as a result of processing limitations that are being actively investigated. In the following paragraphs, we outline candidate processing limitations and lesion characteristics with emphasis on the regulation of DNA end resection.

Several studies and excellent reviews have emphasized IR quality, DNA end structures, DSB complexity, and cell cycle stage, as key parameters, modulating the repair pathway choice after the exposure of cells to IR [4,137,138,139,140]. Among the listed parameters, the cell cycle phase has been recognized as a key determinant of DSB repair pathway selection. It has been shown in several reports that the fluctuations in the protein levels of key DNA damage signaling and repair factors can affect the normal cell cycle progression [141,142,143] and thus the repair outcome. Moreover, these studies show that the chemical or genetic ablation of essential repair factors results in the accumulation of unrepaired DNA lesions, generating the replication stress and chronic activation of DNA damage response. Therefore, it can be speculated that the accumulation of cells in a specific phase of the cell cycle combined with impaired DNA damage signaling can also lead to a shift in the balance between DSB repair pathway utilization.

In addition, the accessibility of particular repair factors to DSB is another determinant of the repair pathway choice, which is particularly evident when DSB repair is investigated in the different chromatin compartments (e.g., eu- or heterochromatin) [144,145,146]. However, in the current review, we highlight the importance of DNA end resection and its control as equally crucial components of DSB repair pathway selection (Figure 2).

Recently, multiple factors have been identified that control the choice between major DSB repair pathways at the level of DNA end resection initiation. In this regard, it is generally accepted that, once resected, a DSB can only be repaired by DNA end resection-dependent mechanisms that block repair by c-NHEJ. The process of DNA end resection is regulated at multiple levels but the most relevant mechanism promoting the initiation of the process is through BARD1/BRCA1-dependent CtIP activation (Figure 2) [147].

Many other molecules that stimulate DNA end resection have also been identified [92,148]. A recent study recognizes BMI-1, a subunit of Polycomb repressive complex 1 (PRC1), as a factor promoting HR-mediated DSB repair by enhancing the efficiency of DNA end resection. Mechanistically, BMI-1 mediates DNA end resection by facilitating the recruitment of CtIP to damage sites [149]. Histone lysine demethylase PHF2 has also been identified as a novel regulator of DSB repair pathway choice at the end resection level, acting by assisting the localization of CtIP to DSBs [150]. Additionally, PAB-dependent poly(A)-specific ribonuclease subunit 2 (USP52) directly interacts with CtIP and catalyzes its deubiquitination, thus promoting end resection and HR—mainly by allowing CtIP phosphorylation at T847 [151]. An interesting study further reports that the mismatch repair complex MSH2-MSH3 interacts with a chromatin remodeling protein SMARCAD1 to form a complex that enhances EXO1 activity by facilitating its recruitment to DSBs for long-range DNA end resection (Figure 2) [152].

Other data show that DNA end resection can be promoted by suppressing key components of the c-NHEJ repair pathway. A small Cajal body-specific RNA 2 (scaRNA2) protein has been found to promote end resection by inhibiting the interactions between DNA-PKcs and KU70/80 subunits, thus preventing the catalytic activation of the enzyme. As a result, the inhibition of DNA-PKcs by scaRNA2 and its interaction with LINP1 lncRNA, stimulates MRN/CtIP-dependent DNA end resection [153].

In contrast, 53BP1 and its associated accessory factors (RIF1, PTIP, etc.) have been demonstrated to actively suppress DNA end resection in the G_1_-phase and to promote the repair of DSBs by c-NHEJ [154,155]. It is well documented that the repair of DSBs in mammals is coordinated by the ubiquitin-dependent accumulation of 53BP1 at DSB-flanking chromatin. Owing to its ability to limit DNA end processing, 53BP1 is thought to promote c-NHEJ and to suppress HR. It has been shown that silencing 53BP1, or exhausting its capacity to bind damaged chromatin, transforms limited DNA end resection to hyper-resection, which results in a switch from RAD51-promoted error-free gene conversion to a mutagenic SSA, dominated by the activity of RAD52. Thus, the role of 53BP1 at DSBs could be extended from a simple protector of DSBs against unscheduled DNA end resection to a factor that fosters HR fidelity [156].

These findings illuminate the causes and the consequences of synthetic viability acquired through 53BP1 silencing in cells lacking the functional *BRCA1* tumor suppressor gene. In line with these results, a recent study demonstrated that such cells survive DSB assaults at the cost of increased reliance on RAD52-mediated SSA, which fuels the generation of SCAs and reduces genome stability [157]. However, some findings suggest that, when challenged by DSBs, BRCA1- and 53BP1-deficient cells may become hypersensitive to, and be eliminated by, RAD52 inhibition [156].

Recently, an interesting molecular complex consisting of revertability protein 7 homologue (REV7), a known 53BP1-pathway component, and three hitherto uncharacterized proteins: C20orf196 (SHLD1), FAM35A (SHLD2), and CTC-534A2.2 (SHLD3) has been found to interfere with a single-stranded DNA production during DNA end resection (Figure 2) [158]. This complex, termed Shieldin [159], emerged as a strong candidate for the role of ultimate effector of 53BP1-dependent end protection. The Shieldin complex has not known enzymatic activities but promotes many 53BP1-associated processes, such as the protection of DNA ends, c-NHEJ, and immunoglobulin CSR [160]. Consistent with its role in protecting the DNA ends from nucleases, the Shieldin complex interacts with long single-stranded DNA (longer than 60 nt) but not with double-stranded DNA [158,159,161].

Another study reveals the role of HELB as an additional factor antagonizing DNA end resection [162]. HELB acts independently of 53BP1, and is recruited to single-stranded DNA by interactions with RPA. HELB utilizes its 5’-3’- single strand DNA translocase activity to inhibit EXO1- and BLM/DNA2-mediated long-range end resection. An interesting report based on a loss-of-function CRISPR/Cas9 screen to identify factors responsible for the recovery of DNA end resection in BRCA1-deficient cells identifies DYNLL1 as an inhibitor of DNA end resection [149]. DYNLL1 limits the nucleolytic degradation of DNA ends by interacting with the MRN complex, BLM helicase, and DNA2 endonuclease [163]. Additional activities that operate in the regulation of excessive DNA end resection are BRCA1/BARD1 and RAP80. They form a multisubunit complex along with Abraxas that includes the deubiquitinases BRCC36, BRCC45, and MERIT40, all crucial factors for the retention of BRCA1 at the DSB. However, paradoxically, the complex negatively modulates BRCA1-mediated HR function, counteracts illegitimate HR, and suppresses chromosomal instability [164,165].

In the same context, our recent study identifies DNA-PKcs as a conductor of the overall DNA repair program and a major regulator of DNA end resection, a function which is beyond its role in c-NHEJ [71]. Indeed, our results show that all tested DNA-PKcs mutants show hyper-resection in the G_2_-phase, while mutants with defects in other c-NHEJ activities exhibit levels of resection close to those observed in the parental cell lines—thus ruling out that c-NHEJ suppression underlies these observations. These results challenge the model of simple competition between c-NHEJ and HR, and suggest an integration of DNA-PKcs into the machinery regulating end resection. Our working hypothesis is that DNA-PKcs plays a role in resection regulation and HR by remaining at the break site and assisting DNA end processing. In line with this hypothesis, some studies show a positive correlation between DNA-PKcs activity and DNA end resection [38,166].

In addition to the above factors, a recent study from our group identified the load of DSBs in the genome as a potential determinant of DSB repair pathway choice [157]. The dose-dependent component of the repair pathway choice regulation is outlined in more detail next.

## 4. Repair Pathway Choice by Gauging DSB Load: Suppression of HR with Increasing IR Doses

As outlined in the preceding sections, whether a specific DSB undergoes processing via c-NHEJ, HR, or alternative repair mechanisms is governed by various factors. Mounting evidence highlights the importance of radiation dose in the regulation of DSB repair pathway choice [157,167]. Interestingly, a study employing pulsed-field gel electrophoresis (PFGE) in mutant chicken cell lines lacking critical HR factors, Rad51, Rad51B, Rad52, and RAD54, allowed key insights into DSB repair regulation: there are no discernible differences in DSB repair kinetics among the investigated cell lines despite the grave HR defects. This observation hinted at the limited contribution of HR to the repair of DSBs when cells are exposed to radiation doses ranging between 20 and 80 Gy, which are typically utilized in PFGE investigations [168].

Moreover, DT-40 cells deficient in both Ku70 and Rad54 showed repair defects similar to those observed in single Ku70 mutants, which additionally suggests that c-NHEJ is the dominant repair pathway at IR doses typically used for PFGE [168]. Similar PFGE experiments conducted with wild-type G_2_-phase-enriched cell lines, where HR activity is at a maximum, demonstrate repair capacity similar to that of exponentially growing cells [53]. The limited impact of HR on DSB repair in these experiments is further validated in experiments testing G_2_-phase cell populations exposed to a highly specific small molecule inhibitor, VE-821, which impedes HR through the abrogation of ATR activity [167]. Furthermore, we have reported that actively growing mouse embryonic fibroblasts (MEF) that are defective in HR repair, process DSBs with an efficiency similar to that of wild-type cells [169]. The same also holds true for Chinese hamster mutant cells with defects in the RAD51 paralog genes, *XRCC2* or *XRCC3*, where even in the G_2_-phase of the cell cycle, DSB repair measured by PFGE shows a similar rate to the repair measured in parental cell lines. All of the above experiments suggest that IR dose may be a determinant of HR utilization, as HR is of course known, to contribute to cell survival and the repair of DSBs when analyzed at lower doses of IR.

To evaluate the contribution of HR to the repair of DSBs at more physiological IR doses, relevant to radiation protection and radiation therapy, we developed a cell-cycle specific immunofluorescence analysis of γ-H2AX foci, an established surrogate marker for DSB formation (Figure 3A) [170,171], in combination with the detection of RAD51 foci, as a marker for ongoing HR (Figure 3B) [172]. Figure 3A,B illustrate an idealized data set for γ-H2AX and RAD51 foci formation after IR exposure, based on our previously generated immunofluorescence analysis in G_2_-phase cells [157], where all DSB repair pathways are fully active [157,167,173]. Our results have shown that the maximum number of γ-H2AX foci is reached early after radiation exposure, irrespectively of the applied dose. Moreover, the number of γ-H2AX increases linearly with increasing the dose of radiation (Figure 3A). In contrast, repair foci formed by RAD51 reach a maximum, at times increasing with increasing the IR dose, and the number of RAD51 foci fails to increase linearly with increasing radiation dose, but bends toward a plateau above 2 Gy (Figure 3B). This response indicates a dose-dependent component of HR engagement that has not been previously described by other groups, and suggests that HR is actively suppressed at doses where a high DSB load is present in the genome.

By calculating the ratio between RAD51 and γ-H2AX foci, we could evaluate the fraction of DSB engaging HR at any given IR dose (Figure 3C). The generated data demonstrate that at low DSB-loads, when about 20–25 DSBs are formed in a G_2_-phase cell, approximately 50% of the DSBs are repaired by HR. Notably, its contribution decreases to 30% at 50 DSBs per genome, whereas at high DSB-loads (above 200 DSBs), the contribution of HR becomes practically undetectable (Figure 3C). Our study also demonstrated an abundance of RAD51 and other HR-related enzymatic activities (MRE11 and RPA) at high radiation doses, thus ruling out protein depletion as the cause for the effect—as has been previously shown for RPA [174].

Results generated during the above study also support the notion that the suppression of HR activity at high IR doses is a regulated process, delimited but not defined by the 53BP1 and RAD52 activities [157]. The suppression of HR at high IR doses is not an isolated phenomenon, as a separate study utilizing a panel of somatic and lymphoblastic cell lines also demonstrated a pronounced suppression of HR at doses above 2 Gy [167]. Our results also indicated that the process of DNA end resection is uncoupled from HR in terms of IR dose-dependent regulation, as RPA foci formation saturates at much higher doses (around 16 Gy). The excess of DNA end resection under these conditions promotes SSA [157].

The increased contribution of HR at low DSB-loads is confirmed by another study following chromatid break repair as a function of IR dose [175]. When G_2_-phase cells are exposed to IR, approximately 10% of the induced DSBs manifest as chromatid breaks, visualized during mitosis [175]. This phenomenon allows the cytogenetic analysis of the processing of the specific subset of DSBs underpinning chromatid breaks. Notably, at low radiation doses, chromatid breaks are almost exclusively processed by HR. Strikingly, G_2_-phase-specific experiments with wild-type CHO cells exposed to a broad spectrum of radiation doses demonstrated a pathway switch to c-NHEJ with increasing IR dose.

These results further consolidate the utilization of HR at low doses and demonstrate switching to c-NHEJ with increasing dose [175]. Another intriguing piece of evidence that HR is the preferred repair pathway at low IR doses comes from the observation that HR deficient cells are not able to initiate proper DDR and to activate the G_2_-phase checkpoint after exposure to IR doses below 4 Gy [176]. This observation, as well as the recently reported strong ATR dependence of the IR-induced G_2_-phase arrest [173,177,178], strongly supports the connection between checkpoint activation and HR efficiency and identifies HR as the process, mainly benefiting from the checkpoint at low doses in the G_2_-phase of the cell cycle.

## 5. Exploiting Weaknesses of DSB Processing to Improve Radiation Therapy

Defects in gene-encoding proteins implicated in the repair of DSBs by HR are frequently found in various cancers. Numerous studies have demonstrated that germline mutations in *BRCA1*- and *BRCA2*- tumor suppressor genes increase the lifetime risk of breast or ovarian cancer development [179]. Mutations in *BRCA1/2* genes have also been found in many sporadic tumors including pancreatic [180] and prostate cancer [181]. In addition to *BRCA1/2* gene alterations, many other genes involved in HR repair such as *RAD51B*, *RAD51C*, *RAD51D*, *PALB2*, and *BRIP1*, are mutated in many cancer types. Mutational signatures associated with moderate to severe HR deficiency (HRD) also include alterations in the RAD50 and the BLM helicase [182]. Counterintuitively, RAD51 is frequently found overexpressed and has been associated with poor prognosis in patients with solid tumors, thus potentially acting as a driver of illegitimate HR.

On the other hand, tumor types with defects in c-NHEJ are not very common, although they are occasionally found in ovarian [183] and glioblastoma tumors [184]. Components of c-NHEJ are indeed suggested targets in synthetic lethality approaches designed to increase the effects of IR on HR-deficient tumors [185]. One of the most explored targets along these lines is DNA-PKcs, as the abrogation of DNA-PKcs activity by small molecule inhibitors triggers apoptosis in ATM-defective cells, both in vitro and in vivo [186]. The most potent DNA-PKcs inhibitors, M3814, CC-115, and CC-122 are currently being investigated in several clinical trials [187]. Together with ATM, ATR is also a preferred target in combination with a mitigation of DNA-PKcs activity. It has been reported that a combination of KU-0060648, a potent DNA-PKcs inhibitor, with ATR inhibitor, AZD6738 potentiates radiosensitization of head and neck squamous carcinoma cell lines [188].

The BRCA1 and BRCA2 proteins have essential physiological roles in maintaining genome stability in rapidly proliferating tumor cells, not only by promoting DSB-repair via HR but also by facilitating DNA replication. The latter is based on the capacity of BRCA1/2 to restart, stabilize, and protect stalled replication forks against nucleolytic degradation. Therefore, loss of either BRCA1 or BRCA2 activity results in slow rates of replication fork progression [143] and a high frequency of stalled replication forks that are susceptible to degradation by cellular nucleases [189]. Indeed, several reports provide evidence that HR-defective cells exhibit the abnormal accumulation of ssDNA as Okazaki fragments during DNA replication. This phenomenon has been originally observed during DNA replication in the absence of RAD51 [190] and subsequently confirmed under BRCA1/2-defective conditions [191]. The function of HR proteins in DNA replication and the accumulation of persistent ssDNA gaps in HR-deficient tumors have been linked to PARP inhibitor sensitivity and platinum compounds-induced cell killing [192]. Various aspects of PARPi sensitivity and acquired resistance in cancer therapy are reviewed elsewhere [193,194,195].

When arrested, DNA replication forks are extensively degraded and fail to restart, resulting in replisome dissemblance and forks collapse, thus generating DSBs [196]. In the context of *BRCA1/2* deficiency and HR abrogation, replication-associated DSBs rely on the remaining highly efficient DSB repair pathways (c-NHEJ and alt-EJ) for joining the DNA ends. This model also defines the synthetic lethality of PARP inhibition in HR-deficient cells [197]. However, the whole genome sequencing of HR-defective tumors has uncovered that multiple DNA rearrangements are partially due to POLθ-dependent alt-EJ [15]. In various cancers, HR-deficiency is frequently associated with POLθ-overexpression [198,199]. The enhanced engagement of POLθ-dependent alt-EJ represents a compensatory mechanism to complement defective HR-mediated DSB repair. Consistent with this, the inactivation of POLθ-encoding gene *POLQ* is synthetically lethal with defects in genes controlling HR [200,201].

Interestingly, a recent study revealed that, in addition to repairing DSBs, POLθ also counteracts their formation by protecting replication forks from breakage, specifically at the lagging strand during Okazaki fragment synthesis [202]. This function might limit DSB accumulation in HR-deficient, and, to some extent, in HR-proficient, cells, explaining the synthetic lethality induced by the loss of POLθ and either HR [198,201,202] or c-NHEJ [203] defects.

Moderate expression in normal tissue compared with overexpression in a number of cancer types defines POLθ as a promising therapeutic target. It provides strong rationale for the development of highly specific inhibitors for POLθ pharmacological inactivation. Recently, Artios Pharma has announced the development of the first selective, orally bioavailable, small molecule inhibitor of the POLθ polymerase domain, ART4215. The inhibitor was approved by the U.S. Food and Drug Administration (FDA) in July 2021 and is currently in phase I/IIa clinical trials for the treatment of patients with advanced or metastatic solid tumors—as a monotherapy or in combination with talazoparib or niraparib.

BRCA1/2 mutations in PARP-inhibitor-resistant cells often display a TMEJ-specific mutational signature, hence it is possible that POLθ might contribute to the acquisition of PARP inhibitor resistance [204]. In addition, resistance to PARP inhibitors can occur via the loss of 53BP1, a gene shown to be synthetically lethal with POLθ, thereby rendering these cells dependent on POLθ [205,206]. As a consequence, using POLθ inhibitors in combination with PARP inhibitors, or as a second-line therapy, might prolong drug response and suppress resistance acquisition [207]. It remains to be elucidated whether POLθ inhibition will also be beneficial for the treatment of cells that acquire PARP inhibitor resistance via other mechanisms, such as the loss of the Shieldin complex [161]. Furthermore, pharmacological targeting of major factors orchestrating DNA-end resection, as well as of RAD51 recombinase offers attractive possibilities to suppress HR and generate PARP1 sensitivity [28,208].

Finally, it is widely accepted that genetic ablation or the targeted inhibition of HR repair proteins offer a wide range of possible applications in cancer therapy with great potential in radiotherapy as well. The increased contribution of HR at low doses of IR significantly strengthens this rationale, as doses in the range of 2 Gy are routinely used in the clinic. Moreover, combination therapy utilizing IR and some of the inhibitors described above has the potential to further improve cancer management by IR.

## 6. Conclusions

The accumulated results in the literature and our recent study shed light on DSB repair processes at IR doses, relevant to radiation protection and radiation therapy. They point to DSB repair pathways that may be targeted with inhibitors to enhance tumor radiosensitization and provide a rationale for searching for new molecular targets and for developing novel DSB repair pathway-specific inhibitors. Cancer management by radiation aims to maximize tumor response, while minimizing side effects on healthy tissue, often in the form of fractionated radiotherapy. The reported dose-dependent regulation of DSB repair pathways choice might provide a rationale for certain forms of fractionated radiation therapy.

## Figures and Tables

**Figure 1 ijms-24-14956-f001:**
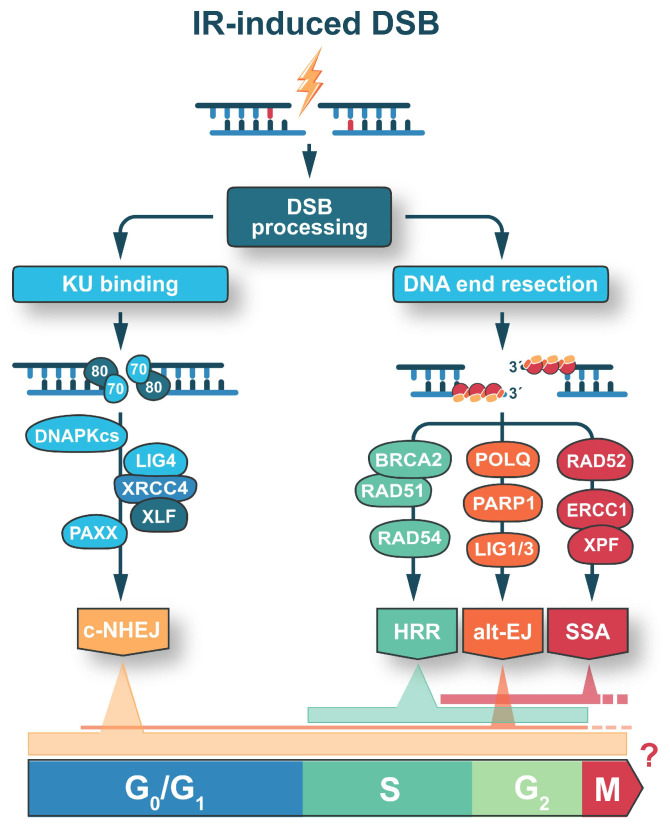
DSB repair pathways and their activity throughout the cell cycle. DNA end resection is the main delimiter of DSB repair pathway engagement. Thus, DSB repair pathways are categorized as DNA end resection independent (c-NHEJ) and DNA end resection dependent (HR, alt-EJ, and SSA). C-NHEJ operates with similar efficiency throughout the entire cell cycle, while HR, alt-EJ, and SSA are more active in S- and G_2_-phase, where resection activities are typically higher.

**Figure 2 ijms-24-14956-f002:**
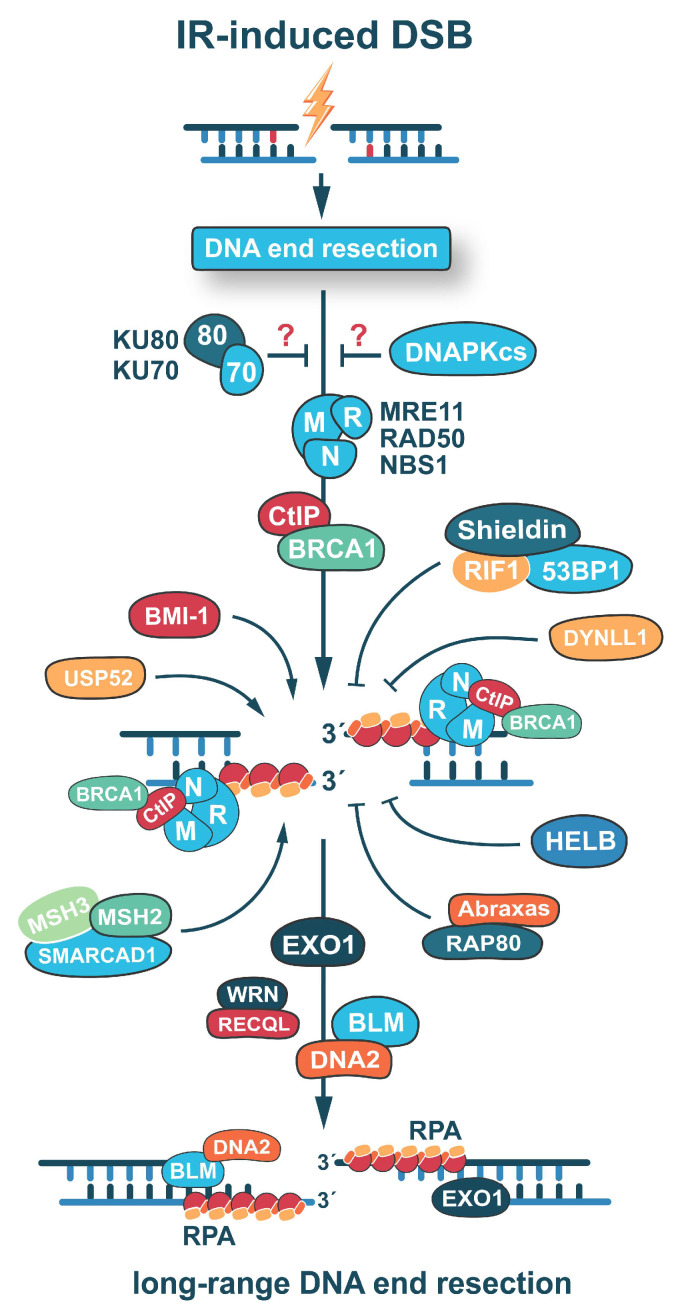
Proteins involved in the regulation of DNA end resection. DNA end resection is initiated by DSB recognition through the MRN complex, comprised of MRE11, RAD50, and NBS1, which subsequently recruits BRCA1-activated CtIP to initiate short-range DNA end resection. The c-NHEJ factors, KU70/80 and DNA-PKcs, are thought to inhibit DNA end resection. Multiple factors regulate CtIP recruitment and activation at DSBs. Newly identified positive and negative regulators of DNA end resection are depicted. Long-range resection is initiated by the recruitment of EXO1 and BLM/DNA2 activities and may be further promoted by the WRN/RECQL complex. Long-range resection results in the generation of long single strand 3′-overhangs, coated by RPA, that promote RAD51 nucleoprotein filament formation and HR. Accidents in the initiation of long-range resection or RAD51 filament formation may cause shifts in DSB repair to mutagenic pathways.

**Figure 3 ijms-24-14956-f003:**
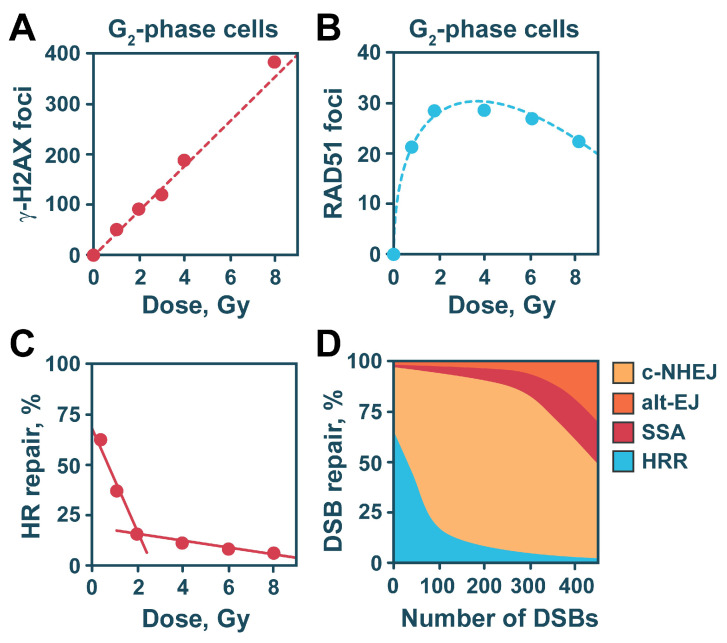
Dose-dependent regulation of DSB repair pathway choice. (**A**,**B**) Idealized dose response curves of *γ*-H2AX and RAD51 foci formation depicting the linear increase of γ-H2AX foci with increasing radiation dose and the saturation of RAD51 foci at higher IR-doses. The plots show fictive data points based on previously published results. (**C**) The numbers of RAD51 and γ-H2AX foci are used to calculate their ratio that indicates the fraction of DSBs processed by HR (HR repair, %). It is evident that HR contributes more to DSB repair at low IR doses, while its contribution at high doses is reduced. (**D**) Diagram showing estimates of the relative involvement of the different DSB repair pathways with increasing DSB-load. It is evident that c-NHEJ is the dominant repair pathway at medium and high DSB-loads, while HR is engaged at low DSB loads. Alternative forms of DSB repair gain ground when HR is suppressed at high doses. Moreover, under conditions of excessive DNA end resection (high-DSB load), SSA is promoted and significantly contributes to the repair of DSBs.

## Data Availability

Not applicable.

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
