# Peer review of "New Facets of DNA Double Strand Break Repair: Radiation Dose as Key Determinant of HR versus c-NHEJ Engagement"

_ijms, 2023, doi:10.3390/ijms241914956_

Round 1

Reviewer 1 Report

Mladenov et al’s manuscript is entitled as “New facets of DNA double strand break repair: This is an interesting review, which emphasises new advancements in the field of DSB repair, especially ionizing radiation induced DSB generation and repair. In the review, authors have brought out several interesting aspects:  (1) functional interplay between major DSB repair pathways in combination with their cell cycle requirements (2) Pathway choices and (3) therapeutic implications. Overall, the manuscript is well written and provides new insights into the DSB repair process based on several interesting literatures. Authors may address following concerns to enhance the quality of the review and better understanding of the new readers in the field 

1.     Line 242: “Initially, CtIP generates a nick in one of the DNA strands near the 5′- end and 242 initiates MRE11 dependent 3′- to 5′-nucleolytic degradation to create short single stranded 3′-overhangs [85, 86]. The short-range resection is followed by long range resection mediated by the recruitment of multiple nucleases like Bloom syndrome helicase/DNA replication ATP-dependent helicase/nuclease 2 (BLM/DNA2) complex and exonuclease 1 (EXO1) (Figure 2).” ……………Apart from BLM/DNA2/EXO1, WRN RECQL helicase also known to play important role long patch resections. Depletion of CTIP and WRN together is reported to have an severe effect on   DSB resections in IR treated cells – Authors should include some of the these recent reports to highlight important roles of these proteins as they essentially modulate the fate of the repair DSB repair process differently in cancer cells.

2.     Authors may give figures to depict alt-EJ and SSA and mention proteins involved in the process .    

3.     Section: “Determinants of DSB repair pathway choice”. This is an interesting aspect of DSB repair and authors have nicely emphasised the recent advancements in this area. Since the review is focussing upon IR related DSB repair, authors are suggested to include some of the recent literature for pathway choice in response to IR treatment.

4.     Figure 3A: this is quite interesting to correlate the contribution of NHEJ and HR with IR dose. Is the figure shown is unpublished data of authors or from literature? This needs to be clearly mentioned in the text part (Line 485-491)

5.     Line 509 “In aggregate our findings summarized in Figure 3D allows to elaborate on DSB repair pathway utilization at different DSB load.” In aggregate should be replaced with “Together” or some better phrase.

6.     Line 586: reference is missing

7.     Line 620: In aggregate may be replaced

8.     Authors have discussed predominant role of NHEJ at higher does and role of both NHEJ and HR at lower IR dose. This is quite interesting radiation therapy point of views. It would be also nice to give some perspectives for temporal relationship of DSB repair process viz: it is seen that although most of the DSBs are repaired within few hours of radiation, few residual DSBs persist for longer time (> 24 h). It seems these residual DSBs are engaged with RAD51/HR factors.     

Author Response

Response to Reviewers’ comments

 IJMS

“New facets of DNA double strand break repair”

Mladenov et al.

ijms-2597854

Response to the Review Report 1

“Mladenov et al’s manuscript is entitled as “New facets of DNA double strand break repair: This is an interesting review, which emphasizes new advancements in the field of DSB repair, especially ionizing radiation induced DSB generation and repair. In the review, authors have brought out several interesting aspects: (1) functional interplay between major DSB repair pathways in combination with their cell cycle requirements (2) Pathway choices and (3) therapeutic implications. Overall, the manuscript is well written and provides new insights into the DSB repair process based on several interesting literatures. Authors may address following concerns to enhance the quality of the review and better understanding of the new readers in the field.”

We thank the reviewer for highlighting the essence of the current review article and for recognizing its potential contribution to the field. We also appreciate the constructive criticism and the suggestions for improving the manuscript.

In the revised version of the manuscript, we made every effort to address the points and concerns raised by the reviewer and explain in detail the rationale of our response. We hope that by doing so, we improved the manuscript and enhanced its clarity.

Major comments:

“1) Line 242: “Initially, CtIP generates a nick in one of the DNA strands near the 5′- end and initiates MRE11 dependent 3′- to 5′-nucleolytic degradation to create short single stranded 3′-overhangs [85, 86]. The short-range resection is followed by long range resection mediated by the recruitment of multiple nucleases like Bloom syndrome helicase/DNA replication ATP-dependent helicase/nuclease 2 (BLM/DNA2) complex and exonuclease 1 (EXO1) (Figure 2).” Apart from BLM/DNA2/EXO1, WRN RECQL helicase also known to play important role long patch resections. Depletion of CTIP and WRN together is reported to have an severe effect on DSB resections in IR treated cells – Authors should include some of the these recent reports to highlight important roles of these proteins as they essentially modulate the fate of the DSB repair process differently in cancer cells.

We thank the reviewer for highlighting the significance of WRN/RECQL axis for the processive DNA end resection. In the revised version of the manuscript, we have implemented the reviewer´s suggestion by indicating some of the recent findings about the role of WRN and RECQL helicases in DNA end resection.

“2) Authors may give figures to depict alt-EJ and SSA and mention proteins involved in the process.”

We highly appreciate the suggestion. However, since the focus of the article is on resection and the interplay between c-NHEJ and HR with increasing radiation dose, we refrained from adding detailed figures for each individual pathway, including alt-EJ. Actually, general aspects of DSB repair pathways are illustrated in Figure 2. We also provide references to other reviews for the interested reader.

“3) Section: “Determinants of DSB repair pathway choice”. This is an interesting aspect of DSB repair and authors have nicely emphasized the recent advancements in this area. Since the review is focusing upon IR related DSB repair, authors are suggested to include some of the recent literature for pathway choice in response to IR treatment.”

We thank the reviewer for this meaningful suggestion. We have extended the indicated section with references highlighting recently reported determinants of DSB repair pathway choice after IR exposure.

“4) Figure 3A: this is quite interesting to correlate the contribution of NHEJ and HR with IR dose. Is the figure shown unpublished data of authors or from literature? This needs to be clearly mentioned in the text part (Line 485-491).”

The figure reflects idealized results drawn based on our published data (Mladenov et al., Strong suppression of gene conversion with increasing DNA double-strand break load delimited by 53BP1 and RAD52. Nucleic Acids Res 2020, 48, 1905-1924). We clarify this in the revision.

“5) Line 509 “In aggregate our findings summarized in Figure 3D allows to elaborate on DSB repair pathway utilization at different DSB load.” In aggregate should be replaced with “Together” or some better phrase.”

We agree and corrected the expression.

 “6) Line 586: reference is missing”

The missing reference has been added.

“7) Line 620: In aggregate may be replaced”

Suggestion considered as above.

“8) Authors have discussed predominant role of NHEJ at higher does and role of both NHEJ and HR at lower IR dose. This is quite interesting radiation therapy point of views. It would be also nice to give some perspectives for temporal relationship of DSB repair process viz: it is seen that although most of the DSBs are repaired within few hours of radiation, few residual DSBs persist for longer time (> 24 h). It seems these residual DSBs are engaged with RAD51/HR factors.”

The Review focuses on the analysis of processing of the bulk of DSBs. The processing and characteristics of very small subsets of DSBs escaping standard processing and remaining for longer times fall outside this scope – despite their potential importance in cell killing. We have extended the Conclusions section to make this point, but refrained from extensively reviewing available information along the lines indicated by the Reviewer, although we discuss temporal aspects of RAD51 foci. In the revised manuscript, we also point out the translational significance of the reviewed information and provide clinically relevant perspectives.

Reviewer 2 Report

Review of manuscript ijms-2597854 - "New facets of DNA double-strand break repair"

This manuscript presents a review of four major pathways involved in the repair of DNA double-strand breaks and emphasizes their relevance to cancer therapy, especially radiation therapy. It appears to be an excellent review. There are a few mistakes that should be corrected and there are also issues regarding sentence grammar that should be addressed.

Comments:

1. Page 4, line 173 - "...DNA-PKcs activity is detected in invertebrates, fungi, plants, and protists as well [67]." My understanding is that detection in the study referred to here was based on sequence and structural similarities rather than enzyme activity.

It may be better to write: "...DNA-PKcs-specific sequence and structural motifs were detected in invertebrates, fungi, plants,...".

2. Page 8, line 344 - "The evolution of four, and not just one or two as is often the case, mechanistically distinct DSB repair pathways strongly suggests that the preservation of genome integrity is of paramount importance in eukaryotic cells." The underlined 2nd half doesn't logically follow from the beginning part because the focus of the four mechanisms is on DSBs specifically. It may be better to say "...DSB repair pathways is an indication of the deleterious nature of DSBs and how important it is to repair them efficiently."

Page 8, line 347 - "...but with progressively diminishing..." implies a time element and probably should be changed. Perhaps something like "...even though three of them have reduced fidelity..."

Page 11, line 509 - "...summarized in Figure 3D...". There is no Figure 3D.

Page 12, Figure 3 - The graphs should be labelled as A,B,C and D. Also, the top two graphs do not have numbers on the x-axes like the one labelled as "B" and numbers should be included.

Page 9 and beyond - The authors discuss the consequences of suppression or inactivation of specific  proteins affecting repair and replication in various places in the text. One phenotype that I did not see mentioned was that reduction of these functions can lead to chronic activation of the DNA damage checkpoint stress response. The idea is that increased levels of unrepaired spontaneous DNA damage in the repair-deficient cells rises above a threshold and induces changes/arrest in cycling. Examples of such studies are Dodson, Shi and Tibbetts, JBC 2004, Mukherji, Bell, Supekova, et al. PNAS 2006, and Maya-Mendoza Moudry, Merchut-Maya et al. Nature 2018, and others. Note that I have no connection to any of the authors of the aforementioned studies.

All pages - The text of the manuscript needs additional editing to fix many of the sentences. The primary problems are related to the following:

- many sentences use present tense but should be in past tense, especially when referring to the results of a previous study, e.g., results "are shown" should be "were shown", etc.

- leaving out or including inappropriately "a", "an" and "the" before words,

- inconsistent use of hyphens, e.g., single-stranded, double-stranded, resection-dependent, resection-independent, etc.

Minor comments:

Page 2, line 66 - mechanisms of DSBs repair should be mechanisms of DSB repair

Page 3, line 110 - compromised  should probably be changed to reduced or decreased

Page 5, line 229 - build-in should be built-in

Page 7, line 299 - RAN50 should be RAD50

Page 9, line 359 - "...ablation of anyone results in genomic instability" should be "...ablation of any one of them results in genomic instability"

Page 9, line 379 - ...end resection are also identified [87, 134] should be ...end resection have also been identified [87, 134]

Page 14, line 628 - "It identifies repair pathways..." should be "It has identified repair pathways..."

As mentioned in the review file, there needs to be some modest editing to fix grammar and tense problems in sentences.

Author Response

Response to Reviewers’ comments

 IJMS

“New facets of DNA double strand break repair”

Mladenov et al.

ijms-2597854

Response to the Review Report 2

“This manuscript presents a review of four major pathways involved in the repair of DNA double-strand breaks and emphasizes their relevance to cancer therapy, especially radiation therapy. It appears to be an excellent review. There are a few mistakes that should be corrected and there are also issues regarding sentence grammar that should be addressed.”

We highly appreciate the reviewer's positive evaluation. In the revised version, we addressed all issues raised and edited the manuscript to remove indicated weaknesses.

Comments:

“1) Page 4, line 173 - "...DNA-PKcs activity is detected in invertebrates, fungi, plants, and protists as well [67]." My understanding is that detection in the study referred to here was based on sequence and structural similarities rather than enzyme activity.

It may be better to write: "...DNA-PKcs-specific sequence and structural motifs were detected in invertebrates, fungi, plants,...".”

We agree and have corrected the passage.

“2) Page 8, line 344 - "The evolution of four, and not just one or two as is often the case, mechanistically distinct DSB repair pathways strongly suggests that the preservation of genome integrity is of paramount importance in eukaryotic cells." The underlined 2nd half doesn't logically follow from the beginning part because the focus of the four mechanisms is on DSBs specifically. It may be better to say "...DSB repair pathways is an indication of the deleterious nature of DSBs and how important it is to repair them efficiently…””

Again, we agree and rephrased the passage.

“3) Page 8, line 347 - "...but with progressively diminishing..." implies a time element and probably should be changed. Perhaps something like "...even though three of them have reduced fidelity..."”

Again, we agree and adopted the suggestion.

“4) Page 11, line 509 - "...summarized in Figure 3D...". There is no Figure 3D”

We apologize for the mistake. Figure 3 is now correctly annotated.

“5) Page 12, Figure 3 - The graphs should be labelled as A,B,C and D. Also, the top two graphs do not have numbers on the x-axes like the one labelled as "B" and numbers should be included.”

We have corrected the indicated deficiencies and errors.

“6) Page 9 and beyond - The authors discuss the consequences of suppression or inactivation of specific proteins affecting repair and replication in various places in the text. One phenotype that I did not see mentioned was that reduction of these functions can lead to chronic activation of the DNA damage checkpoint stress response. The idea is that increased levels of unrepaired spontaneous DNA damage in the repair-deficient cells rises above a threshold and induces changes/arrest in cycling. Examples of such studies are Dodson, Shi and Tibbetts, JBC 2004, Mukherji, Bell, Supekova, et al. PNAS 2006, and Maya-Mendoza Moudry, Merchut-Maya et al. Nature 2018, and others. Note that I have no connection to any of the authors of the aforementioned studies.”

This is a great suggestion, and we thank the reviewer for pointing-out this. In the revised version of the manuscript, we have added a paragraph including the suggested information and references.

“7) All pages - The text of the manuscript needs additional editing to fix many of the sentences. The primary problems are related to the following:

- many sentences use present tense but should be in past tense, especially when referring to the results of a previous study, e.g., results "are shown" should be "were shown", etc.

- leaving out or including inappropriately "a", "an" and "the" before words,

- inconsistent use of hyphens, e.g., single-stranded, double-stranded, resection-dependent, resection-independent, etc.”.

We carefully edited the manuscript in an effort to reduce some of the indicated deficiencies.

Minor Comments:

 “8) Page 2, line 66 - mechanisms of DSBs repair should be mechanisms of DSB repair

Corrected.

“9) Page 3, line 110 - compromised should probably be changed to reduced or decreased”

Corrected.

“10) Page 5, line 229 - build-in should be built-in”

Corrected.

“11) Page 7, line 299 - RAN50 should be RAD50”

Corrected.

“12) Page 9, line 359 - "...ablation of anyone results in genomic instability" should be "...ablation of any one of them results in genomic instability"”

Corrected.

“13) Page 9, line 379 - ...end resection are also identified [87, 134] should be ...end resection have also been identified [87, 134]”

Corrected.

“14) Page 14, line 628 - "It identifies repair pathways..." should be "It has identified repair pathways..."”

Corrected.

Round 2

Reviewer 1 Report

Authors have addressed my concerns